# The propagation of economic impacts through supply chains: The case of a mega-city lockdown to prevent the spread of COVID-19

Hiroyasu Inoue[1]*, Yasuyuki Todo[2]

**1** Graduate School of Simulation Studies, University of Hyogo, Kobe, Japan, **2** Graduate School of Economics, Waseda University, Tokyo, Japan

* inoue@sim.u-hyogo.ac.jp

**Data Availability Statement:** Our data is third party data. Through Tokyo Shoko Research described below, anyone can access the same data as the authors. Our data for production network is based on a survey done by Tokyo Shoko Research.

## Abstract

This study quantifies the economic effect of a possible lockdown of Tokyo to prevent the spread of COVID-19. The negative effect of such a lockdown may propagate to other regions through supply chains because of supply and demand shortages. Applying an agent-based model to the actual supply chains of nearly 1.6 million firms in Japan, we simulate what would happen to production activities outside Tokyo if production activities that are not essential to citizens' survival in Tokyo were shut down for a certain period. We find that if Tokyo were locked down for a month, the indirect effect on other regions would be twice as large as the direct effect on Tokyo, leading to a total production loss of 27 trillion yen in Japan or 5.2% of the country's annual GDP. Although the production that would be shut down in Tokyo accounts for 21% of the total production in Japan, the lockdown would result in an 86% reduction of the daily production in Japan after one month.

## Introduction

COVID-19, a novel coronavirus disease, has been spreading worldwide. By March 31, 2020, the total number of confirmed cases of COVID-19 reached 775,306, whereas the total number of deaths was 37,083 [1]. To prevent the spread of COVID-19, most countries have implemented unprecedentedly stringent restrictions, such as a shutdown of national borders, limits on public gatherings, and closures of schools, shops, and restaurants.

In some cases, whole cities and regions have been locked down. For example, Wuhan, the epicenter of the novel coronavirus, was locked down from January 23 to March 27, 2020, including the closure of all public transport and all companies not essential to citizens' survival, including manufacturing plants, for most of the period (Reuters, March 11, 2020). Apparently, the lockdown heavily affected Wuhan's economy and its population of 11 million people. Moreover, because Wuhan, known as one of China's "Detroits", is a center of the automobile industry and supplies parts and components of automobiles to domestic and foreign plants, the effect of the lockdown propagated to other regions of China and other countries

The data are not in the public domain, but are commercially available. Information on data access is given below. Provider: Tokyo Shoko Research, Ltd. JA Bldg., 1-3-1 Otemachi, Chiyoda-ku, Tokyo 100-6810, JAPAN. Tel: +81 (0)3-6910-3142. Fax: +81 (0)3-5221-0712. Web: http://www.tsr-net.co.jp/ Database: TSR Company Profile Data File and TSR Business Linkage File. Description: TSR Company Profile Data File are standard data sets that have been provided for many years in local market. The data are based on TSR's reporters' site visit interviews, which is our most frequent data source. TSR Business Linkage File is a list of relational companies (shareholders, investee, customer and supplier). That is clarified business relationships between companies through investigation where TSR's reporters confirm both customers and suppliers for a subject company.

**Funding:** JP18K04615 (awarded to HI) JP18H03642 (awarded to YT) Japan Society for the Promotion of the Science https://www.jsps.go.jp/english/e-grants/ The funders had no role in study design, data collection and analysis, decision to publish, or preparation of the manuscript.

**Competing interests:** The authors have declared that no competing interests exist.

through supply chains. For example, Honda, a Japanese automobile manufacturer that operates plants in Wuhan, reduced the production of automobiles in Japan due to a lack of supplies of parts from China in early March 2020 (Nikkei Newspaper, March 2, 2020).

Recently, many studies have empirically confirmed that economic shocks propagate across regions and countries through supply chains. Some studies applying econometric analysis to firm-level data show that firms linked with suppliers and customers in areas affected by natural disasters exhibit decreased performance [2–5]. Several others have estimated economic impacts of the spread of COVID-19, incorporating propagation through input-output linkages across sectors and countries /citeOECD2020, McKibbin2020, Bonadio2020, Guan2020, McCann2020. For example, McKibbin and Fernando [6] use a hybrid of dynamic stochastic general equilibrium (DSGE) models and computable general equilibrium (CGE) models assuming international and inter-sectoral input-output (IO) linkages and estimate the effect of the spread of COVID-19 on the world GDP. Guan and others /citeGuan2020 utilize international IO tables at the country-sector level and estimate impacts of lockdowns of heavily infected countries on production in the world economy. According to their estimates, if Europe and the United States imposed a lockdown strategy with a strictness measure of 80% (i.e., 80% of production in most non-essential sectors are shut down) for 2 months, the world value added would decline by 27%.

By contrast, Inoue and Todo [7] apply firm-level data for the actual supply chains in Japan, rather than inter-sectoral IO tables, to an agent-based model to simulate the production trajectory after the Great East Japan earthquake in 2011. Their results indicate that the propagation effect of an economic shock through supply chains was substantially larger than the direct effect of the earthquake. Inoue and Todo [7] also show that complex network characteristics of supply chains, such as scale-free properties and complex loops, aggravate the propagation effect. Without any network complexity, i.e., if they assume no firm-level inter-linkages but only inter-industry linkages or assume a randomly determined network with no complexity, they find that the propagation effect is quite small.

The analytical framework of Inoue and Todo [7] has several advantages. First, although some studies quantitatively find that supply chains are equipped with complex network characteristics /citeborgatti2009social, battini2007towards, Fujiwara10, they do not examine propagation of shocks through supply chains. Second, the conclusion of Inoue and Todo [7] that the structure of networks significantly influences propagation is consistent with recent findings in the network science literature [8–13]. For example, diffusion is generally promoted when the degree distribution follows a power law and, hence, there are several hub nodes with a large number of links. In such a network, when a shock reaches a hub node, it spreads quickly to many others. Third, although the econometric analysis of /citeBarrot2016, Boehm2019, Carvalho2016, Kashiwagi2018 quantitatively confirms propagation through supply chains, their analysis cannot estimate the total effect of a shock that can be obtained from the framework of Inoue and Todo [7]. Finally, the studies estimating the effect of COVID-19 rely on either a macroeconomic econometric model at the country level [14] or a general equilibrium model assuming international and inter-sectoral IO linkages [6, 15–17] and thus do not incorporate complex inter-firm linkages. As a result, the estimates of the previous studies may be largely undervalued, as suggested by the findings of Inoue and Todo [7].

Therefore, this study utilizes the framework of Inoue and Todo [7] and examines propagation of the economic effect of lockdown of a city to prevent the spread of COVID-19 to other regions through supply chains. In particular, when a large industrial city connected with other regions through supply chains in a complex manner is locked down, the propagation effect may be quite large. Therefore, we focus on the economic effect of a lockdown of Tokyo on other regions. Specifically, by applying the agent-based model developed in Inoue and Todo

[7] to actual supply-chain linkages of approximately 1.6 million firms in Japan, we simulate what would happen to production activities outside Tokyo if Tokyo were locked down or non-essential production activities in Tokyo were shut down for a certain period. Tokyo is an appropriate case for the purpose of this study because it is one of the largest cities in the world and a hub for global supply chains and was, in fact, locked down in April and May 2020. Using this framework, we can obtain a reasonable estimate of the propagation effect of lockdown across regions that cannot be computed from other studies using econometric approaches or IO tables and thus can highlight the impact of the propagation through supply chains.

## Data

The data used in this study are taken from the Company Information Database and Company Linkage Database collected by Tokyo Shoko Research (TSR), one of the largest credit research companies in Japan. The former includes information about the attributes of each firm, including the location, the industry, the sales, and the number of employees, whereas the latter includes major customers and suppliers of each firm. Because of data availability, we utilize data for firm attributes and supply chains in 2016. The number of firms in the data is 1,668,567, and the number of supply chain links is 5,943,073. That is, our data identify major supply chains of most firms in Japan, although they lack information about supply chain links with foreign entities. Because the transaction value of each supply chain tie is not available in the data, we estimate sales from a particular supplier to each of its customers and consumers using the total sales of the supplier and its customers and the input-output (IO) tables for Japan for 2015. In this estimation process, we must drop firms without any sales information. Accordingly, the number of firms in our further analysis is 966,627, and the number of links is 3,544,343. Although firms in the TSR data are classified into 1,460 industries according to the Japan Standard Industrial Classification, we simplify them into 187 industries classified in the IO tables. S1 Appendix provides details of the data construction process.

In the supply chain data described above, the degree, or the number of links, of firms follows a power-law distribution (see S1 Fig), as often found in the literature [12]. The average path length between firms, or the number of steps between them through supply chains, is 4.8. This small average path length indicates that the supply chains have a small-world property, i.e., firms are indirectly and closely connected through supply chains. Therefore, we would predict that economic shocks quickly propagate through the supply chains. Using the same dataset, previous studies [7, 18] find that 46-48% of firms are included in the giant strongly connected component (GSCC), in which all firms are indirectly connected to each other through supply chains. The large size of the GSCC prominently shows that the network has numerous cycles and a complex nature, which is unlike the common image of a layered supply chain structure.

## Method

### Model

Our simulation employs the dynamic agent-based model of Inoue and Todo [7, 19], an extension of the model of Hallegatte [20] that assumes supply chains at the firm level. In the model, each firm utilizes inputs purchased from other firms to produce an output and sells it to other firms and consumers. Supply chains are predetermined and do not change over time in the following two respects. First, each firm utilizes a firm-specific set of input varieties and does not change the input set over time. The variety of inputs is determined by the industry of the producer, and hence, firms in the same industry are assumed to produce the same output. Second, each firm is linked with fixed suppliers and customers and cannot be linked with any new firm

over time. Furthermore, we assume that each firm keeps inventories of each input at a level randomly determined from the Poisson distribution. Following Inoue and Todo [7], in which parameter values are calibrated from the case of the Great East Japan earthquake, we assume that firms aim to keep inventories for nine days of production on average.

When a lockdown directly or indirectly causes a reduction in the production of particular firms, the supply of products of these firms to their customer firms declines. Then, one way for customer firms to maintain the current level of production is to use their inventories of inputs. Alternatively, customers can procure input from their other suppliers in the same industry already connected prior to the lockdown if these suppliers have additional production capacity. If the inventories and inputs from substitute suppliers are insufficient, customers have to reduce their production because of a shortage of inputs. In addition, suppliers of the firms directly affected by the lockdown may have to reduce production because of the reduction of demand from affected customers. Accordingly, the economic shock propagates both downstream and upstream through supply chains. S2 Appendix provides the details of the model.

## Simulation procedure

In the simulation, we assume an extreme case of Tokyo's lockdown where all production activities that are not essential to citizens' survival (hereafter referred to as non-essential production activities) in the central part of Tokyo (23 wards, hereafter simply referred to as Tokyo) are shut down for either one day, one week, two weeks, one month, or two months. Essential production activities are defined as those in the wholesale, retail, utility, transport, storage, communication, healthcare, and welfare sectors (see S3 Appendix for details). After the lockdown period, all sectors immediately resume production at the same level as in the pre-lockdown period. Because the inventory target of each firm is randomly sampled from the Poisson distribution (see S2 Appendix), we run thirty simulations with different sets of inventory targets of firms for each lockdown duration and average over the thirty sets of results.

It should be noted that our assumption does not exactly match the experience of Japan. In practice, the Japanese government announced the state of emergency for some prefectures including Tokyo from April 7 to May 25, 2020, and for the whole country from April 16 to May 14. According to the Act on Special Measures for Pandemic Influenza and New Infectious Diseases Preparedness and Response, the Japanese government can declare a state of emergency if the spread of a infectious disease, such as COVID-19, is rapid and nationwide. In the state of emergency, some economic activities were requested to be closed, and people were requested to reduce their contacts with others by 80%. Although there was no legal punishment for disobeying the requests, strong social pressure in Japan led people and businesses to voluntarily restrict their activities to a large extent. As a result, production activities including those in sectors not officially restricted were reported to shrink substantially (Mainichi Newspaper, May 27, 2020). Because it is still unclear to what extent economic activities shrank, we assume a complete shutdown of non-essential production activities. In addition, to highlight the propagation of the effect of a lockdown of a mega-city to other regions, we focus on a lockdown of Tokyo, rather than the entire economy.

## Results

### Benchmark results

When Tokyo is locked down, the value-added production of Tokyo immediately becomes almost zero. Because the daily production of non-essential sectors in Tokyo is estimated to be 309 billion yen, or approximately 2.9 billion US dollars, the total direct loss of production in Tokyo due to the lockdown is 309 billion yen multiplied by the number of days of the

**Table 1. The loss of value added because of Tokyo lockdowns.** This table shows the results from the simulations assuming the shutdown of all non-essential production activities. These results are based on the average of the simulations. (Unit: trillion yen).

|  | Direct effect on Tokyo | Indirect effect on other regions in Japan | Total effect (% of GDP) |
|---|---|---|---|
| 1 day | 0.309 | 0.252 | 0.561 (0.106) |
| 1 week | 2.17 | 1.54 | 3.70 (0.699) |
| 2 weeks | 4.33 | 4.92 | 9.25 (1.75) |
| 1 month | 9.28 | 18.4 | 27.6 (5.22) |
| 2 months | 18.6 | 49.5 | 68.1 (12.8) |

lockdown period. Table 1 shows the direct production loss in Tokyo for each case in the second column. In addition, the third column shows the production loss outside Tokyo (which includes essential sectors in Tokyo; hereafter, this area is simply called outside Tokyo), which is the propagation effect through supply chains. The indirect loss is calculated from a total loss less the direct damage over the duration of the lockdown and the following three weeks. As shown in Fig 1, after the three weeks, the production fully recovers. The total loss is the total gap between the pre-lockdown daily production and the current daily production. The results in Table 1 are the averages of the simulations.

The results indicate that when Tokyo is locked down for only one day, the production loss outside Tokyo, though not locked down, is already 252 billion yen, 82% of the production loss in Tokyo. When the lockdown continues for a month, the indirect effect on other regions is twice as large as the direct effect on Tokyo, and the estimated total production loss is 27.6 trillion yen, or 5.22% of the annual GDP. In Fig 1, each line shows the dynamics of the total daily value added in Japan in each case assuming different lockdown durations. When the lockdown continues for a month, the daily value added of Japan becomes only approximately one-seventh of that before the lockdown. This result implies that even when the initial production loss in Tokyo is small, its propagation effect on other regions can become large as the lockdown continues.

Fig 2 shows temporal and geographical visualizations of the simulation of a lockdown. A video of the visualization is available (see S1 Video). The reductions in firm production are averaged in municipalities. The red areas indicate that the production in the area is less than or equal to 20% of firms' capacity on average, whereas the light red and orange areas show firms with a more moderate decline in production. The left figure illustrates that a non-negligible number of areas (firms) distant from Tokyo are already affected on the first day of the lockdown. Two weeks later, affected areas spread throughout the country, as shown in the right figure. These visualizations indicate that the indirect effect propagates geographically as the lockdown is prolonged.

## Alternative specifications

In addition to the benchmark simulations above, we experiment with three alternative sets of simulations. First, we assume that all production activities including essential activities are shut down. Then, the daily production loss in Tokyo is 471 billion yen, 52% larger than that in the benchmark simulations (309 billion yen). However, we find that the total production loss in Japan from a lockdown of Tokyo for a month is 31.1 trillion yen, only 13% larger than that in the benchmark (27.6 trillion yen) (see S1 Table for details).

Second, we assume that industrial demand is prioritized over consumer demand so that production activities outside Tokyo will be less affected. For example, computers can be used

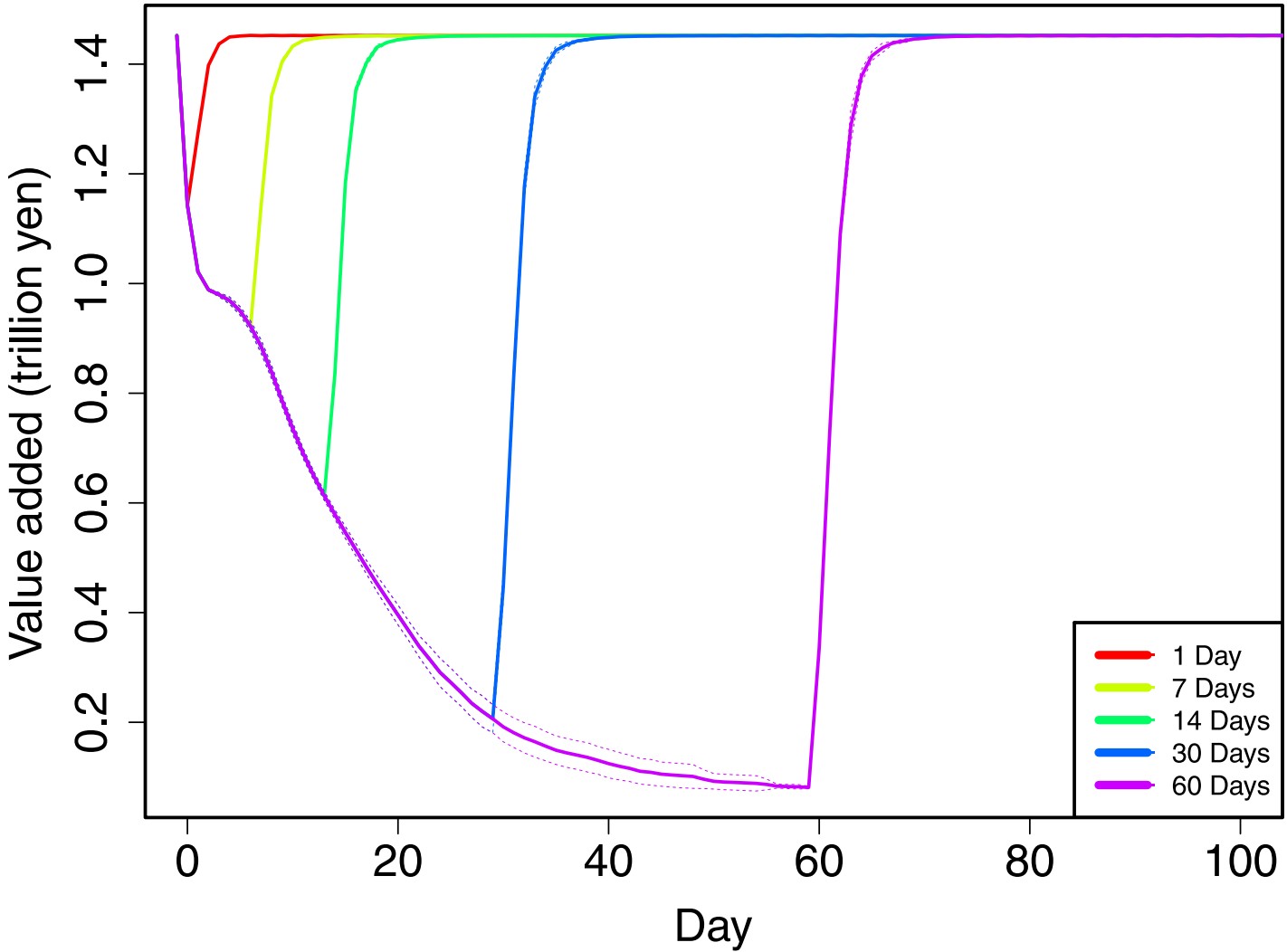

**Fig 1. The dynamics of daily value added in Japan after the lockdown of Tokyo.** Each line shows the average of thirty simulations in which firms have inventory sizes sampled from the Poisson distribution. The dotted lines show the standard deviations. This figure shows simulation results assuming the shutdown of all non-essential production activities.

both by customer firms for production and citizens for consumption. In the benchmark simulation, when the output of a product is not sufficient because of a lockdown, we assume that the limited output is rationed to customer firms and consumers based on their relative demand prior to the lockdown. However, in this alternative simulation, customer firms are prioritized to maximize production in downstream firms (see S2 Appendix for details). Then, we find that a one-month lockdown results in a production loss in Japan of 27.0 trillion yen. Because the production loss does not substantially change from the benchmark result (27.6 trillion yen), we conclude that industry prioritization is not very effective in alleviating the propagation effect of a lockdown.

Finally, we investigate the assumption of a moderate initial shock. Since we only know the addresses of the firms' headquarters, we assign each firm to each headquarters' location in the benchmark and the above alternatives. However, even if the headquarters of a firm is located in Tokyo, other business establishments and factories may continue to operate (however, it may not be possible to obtain such detailed trade information between business establishments

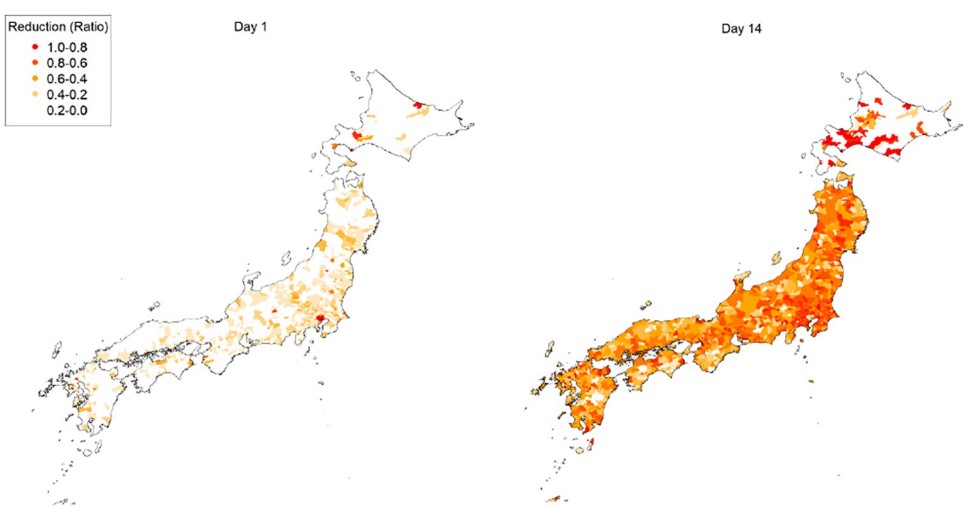

**Fig 2. Temporal and geographical visualizations of the reduction in production.** The left and right panels show the first day and the first 2 weeks of the lockdown, respectively. The reductions are aggregated and averaged over firms in municipalities. The red areas, for example, indicate firms in the areas whose actual production is substantially (more than 80%) smaller than their capacity before the lockdown on average.

and factories). To take this situation into consideration, we proportionally assign the production loss to firms that have headquarters in Tokyo based on the number of headquarters (i.e., counted individually), business establishments, and factories in Tokyo over the number in all areas. This setting may be the other side of the extremely mitigated condition because if the headquarters are closed, the other business establishments and factories probably cannot operate at full capacity. Obviously, this assumption shows a much smaller direct shock (107 billion yen per day) than the benchmark shock (309 billion yen per day). However, if the lockdown is prolonged for a month, the total loss is 22.5 trillion yen, a difference of 5.10 trillion yen from the benchmark. We conclude that this difference is not large enough to change the interpretations of the benchmark results.

## Discussion and conclusion

The simulation results clearly show that the effect of a lockdown of Tokyo quickly propagates to other regions outside Tokyo, leading to a substantial effect on the entire Japanese economy. This conclusion is consistent with the previous finding that economic shocks due to natural disasters propagate to non-disaster regions and result in a large total loss /citeInoue2019. Although the production of non-essential sectors in Tokyo accounts for 21.3% of the total production in Japan, a lockdown of Tokyo for a month would result in a reduction of the daily production in Japan of 86%, or 1.25 trillion yen.

In addition, the effect on other regions becomes progressively larger as the duration of the lockdown becomes longer: when the duration doubles, the production loss more than doubles. In the case of a lockdown for one day, the total loss of value added outside Tokyo is 82% of the loss in Tokyo. However, when the lockdown continues for a month, the loss outside Tokyo is twice as large as the loss in Tokyo. This result implies that the effect of a longer lockdown can reach firms that are "farther" from Tokyo along supply chains in terms of the network.

To alleviate the propagation effect through supply chains, one could limit the shutdown of production activities or prioritize producers' use of goods and services over consumers' use.

However, our results indicate that these measures would not work well, particularly when the duration of a lockdown is long.

Our analysis provides several policy implications. First, because the overall effect of a lockdown of a mega-city on the entire economy is extremely large when we take into account its propagation effect through supply chains, lockdowns should be considered a measure of last resort. Instead, the spread of COVID-19 should be prevented earlier using other means and any lockdown of a mega-city should be avoided. Second, because the total effect of a lockdown progressively increases with its duration, a mega-city lockdown, if it cannot be avoided, should be as short as possible. Policymakers should be aware that policies to alleviate the propagation effect may not work when the lockdown duration is long.

Several caveats of this study should be mentioned. First, we assume that firms cannot find any new suppliers when supplies from their suppliers in Tokyo are disrupted. Second, for simplicity, the model assumes that even service sectors have the inventory mechanism. Finally, the model in this study only considers the dynamics of production and ignores possible changes in prices and wages incorporated in the literature [21, 22]. These assumptions may be too strong, leading to an overestimation of the propagation effect. However, our main result that the propagation effect is substantial would hold.

This research was conducted as part of a project entitled "Dynamics of Economy and Finance from the Economic Network Point of View," undertaken at the Research Institute of Economy, Trade, and Industry (RIETI). This research was also supported by MEXT under the Exploratory Challenges on Post-K Computers Program (Studies of Multi-level Spatiotemporal Simulation of Socioeconomic Phenomena, Macroeconomic Simulations). This research used computational resources of the K computer provided by the RIKEN Advanced Institute for Computational Science through the HPCI System Research project (Project ID: hp190148). The authors thank seminar participants at RIETI.

## Supporting information

**S1 Appendix. Data [23].**
(PDF)

**S2 Appendix. Model.**
(PDF)

**S3 Appendix. Definition of essential industries.**
(PDF)

**S1 Fig. Degree distribution of supply chains in Japan.**
(PNG)

**S2 Fig. Overview of the agent-based model.** Products flow from left to right, whereas orders flow in the opposite direction. The equation numbers correspond to those in S2 Appendix.
(TIF)

**S3 Fig.**
(PNG)

**S1 Video. The address of the video is https://youtu.be/-WTg4pWc9HI.**
(TXT)

**S1 Table. Results from Alternative Specifications: The loss of value added because of a Tokyo lockdown according to alternative specifications.** (Unit: trillion yen).
(PDF)

## Author Contributions

**Conceptualization:** Hiroyasu Inoue, Yasuyuki Todo.

**Investigation:** Hiroyasu Inoue.

**Methodology:** Hiroyasu Inoue.

**Writing – original draft:** Hiroyasu Inoue, Yasuyuki Todo.

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
