## [Decision Letter · Decision Letter 0]

27 Jul 2020

PONE-D-20-16047

The Propagation of the Economic Impact through Supply Chains: The Case of a Mega-City Lockdown against the Spread of COVID-19

PLOS ONE

Dear Dr. Inoue,

Thank you for submitting your manuscript to PLOS ONE. After careful consideration, we feel that it has merit but does not fully meet PLOS ONE’s publication criteria as it currently stands. Therefore, we invite you to submit a revised version of the manuscript that addresses the points raised during the review process.

Authors have provided a theoretically rigorous framework to analyze the influence that COVID-10 brought to the world. Two peer reviewers suggest that further modification and supplement are necessary before fully acception by PLOS ONE, and I hold the same view.

We look forward to receiving your revised manuscript.

Kind regards,

Lizhi Xing, Ph.D

Academic Editor

PLOS ONE

Journal Requirements:

3. We note that Figure 2 in your submission contain map images which may be copyrighted. All PLOS content is published under the Creative Commons Attribution License (CC BY 4.0), which means that the manuscript, images, and Supporting Information files will be freely available online, and any third party is permitted to access, download, copy, distribute, and use these materials in any way, even commercially, with proper attribution. For these reasons, we cannot publish previously copyrighted maps or satellite images created using proprietary data, such as Google software (Google Maps, Street View, and Earth). For more information, see our copyright guidelines: http://journals.plos.org/plosone/s/licenses-and-copyright.

3.1.    You may seek permission from the original copyright holder of Figure 2 to publish the content specifically under the CC BY 4.0 license.

3.2.    If you are unable to obtain permission from the original copyright holder to publish these figures under the CC BY 4.0 license or if the copyright holder’s requirements are incompatible with the CC BY 4.0 license, please either i) remove the figure or ii) supply a replacement figure that complies with the CC BY 4.0 license. Please check copyright information on all replacement figures and update the figure caption with source information. If applicable, please specify in the figure caption text when a figure is similar but not identical to the original image and is therefore for illustrative purposes only.

4. Please ensure that you refer to Figure 3 in your text as, if accepted, production will need this reference to link the reader to the figure.

5. We note you have included a table to which you do not refer in the text of your manuscript. Please ensure that you refer to Table 2 in your text; if accepted, production will need this reference to link the reader to the Table.

Reviewers' comments:

Reviewer's Responses to Questions

**Comments to the Author**

1. Is the manuscript technically sound, and do the data support the conclusions?

Reviewer #1: Partly

Reviewer #2: Yes

2. Has the statistical analysis been performed appropriately and rigorously? 

Reviewer #1: N/A

Reviewer #2: Yes

3. Have the authors made all data underlying the findings in their manuscript fully available?

Reviewer #1: Yes

Reviewer #2: Yes

4. Is the manuscript presented in an intelligible fashion and written in standard English?

Reviewer #1: Yes

Reviewer #2: Yes

5. Review Comments to the Author

Reviewer #1: The Paper is written in well mannered. In introduction section, author (s) are requested to explain the pinpoint background and the objective of the research. This leads to a greater interest by the reader.

Literature review section is not enough to describe the research problem. Author (s) are requested to provide strong literature for present research.

The conclusions should show the novel results with the past situation research for this authors are advices to include the general findings of the research. It is advisable to provide the limitation of this research in a very concise manner.

The overall work done by the author is good.

Reviewer #2: This paper aims to quantify the economic effect of a possible lockdown of Tokyo to prevent the spread of COVID-19 by applying an agent-based model with the actual supply chains of nearly 1.6 million firms in Japan, which is a hot topic and provides a relatively quality paper. Especially in 2020, COVID-19 produces dramatically negative effect on global supply chain and many scholars pay more attention to that. There are some minor comments for the authors.

1. As we know, supply chains whether in a country or in the world is a network. Using network quantitative method to explore supply chains is also a major way. This paper employs the dynamic agent-based model of Inoue and Todo that is an extension of the model of Hallegatte, I would like to see the authors spend more time to introduce the differences between the method mentioned in the paper and network quantitative method.

2. COVID-19 likes a kind of shock or emergency and many scholars have studied policy shock and public emergency bring about impact on economic or social development, so the author need to show relative literatures published to deeply illustrate the paper’s highlights.

3. This study attempts to quantify the economic effect of COVID-19 by taking into account the propagation effect across regions through inter-firm supply chains, why does this paper choose firm-level data? I don’t quite clear to see the main contributions at the end of the introduction.

4. Please explain further what does Lockdown means and what the actual prerequisites of Lockdown are?

5. I would like to see the process of how to match firm-level data with IO industry-level data?

6. PLOS authors have the option to publish the peer review history of their article (what does this mean?). If published, this will include your full peer review and any attached files.

Reviewer #1: No

Reviewer #2: No

---

## [Author Response · Author response to Decision Letter 0]

27 Aug 2020

Please find the attached "Response.pdf."

---

## [Editor Report · Decision Letter 1]

3 Sep 2020

The propagation of economic impacts through supply chains: The case of a mega-city lockdown to prevent the spread of COVID-19

PONE-D-20-16047R1

Dear Dr. Inoue,

We’re pleased to inform you that your manuscript has been judged scientifically suitable for publication and will be formally accepted for publication once it meets all outstanding technical requirements.

Kind regards,

Lizhi Xing, Ph.D

Academic Editor

PLOS ONE
---

## [Editor Report · Acceptance letter]

7 Sep 2020

PONE-D-20-16047R1 

The propagation of economic impacts through supply chains:The case of a mega-city lockdown to prevent the spread ofCOVID-19 

Dear Dr. Inoue:

I'm pleased to inform you that your manuscript has been deemed suitable for publication in PLOS ONE. Congratulations! Your manuscript is now with our production department. 

Kind regards, 

on behalf of

Dr. Lizhi Xing 

Academic Editor

PLOS ONE